# Etiological Factors of Molar Incisor Hypomineralization: A Systematic Review and Meta-Analysis

**DOI:** 10.3390/dj11050111

**Published:** 2023-04-24

**Authors:** María Lilia A. Juárez-López, Leslie Vanessa Salazar-Treto, Beatriz Hernández-Monjaraz, Nelly Molina-Frechero

**Affiliations:** 1Postgraduate and Research Unit, FES Zaragoza, National Autonomous University of Mexico, Mexico City 09230, Mexico; 2Pediatric Dentistry Deparment, FES Zaragoza, National Autonomous University of Mexico, Mexico City 09230, Mexico; 3Metropolitan Autonomous University, Xochimilco Campus, Mexico City 04960, Mexico

**Keywords:** molar incisor hypomineralization, prenatal factors, perinatal factors, postnatal factors, children, developmental defects of enamel

## Abstract

Molar incisor hypomineralization (MIH) is a defect of the dental enamel that predominantly affects first molars and permanent incisors. Identifying the significant risk factors associated with MIH occurrence is essential for the implementation of prevention strategies. The purpose of this systematic review was to determine the etiological factors associated with MIH. A literature search was carried out from six databases until 2022; it covered pre-, peri-, and postnatal etiological factors. The PECOS strategy, PRISMA criteria, and the Newcastle–Ottawa scale were used, and 40 publications were selected for qualitative analysis as well as 25 for meta-analysis. Our results revealed an association between a history of illness during pregnancy (OR 4.03 (95% CI, 1.33–12.16), *p* = 0.01) and low weight at birth (OR 1.23 (95% CI, 1.10–1.38), *p* = 0.0005). Furthermore, general illness in childhood (OR 4.06 (95% CI, 2.03–8.11), *p* = 0.0001), antibiotic use (OR 1.76 (95% CI, 1.31–2.37), *p* = 0.0002), and high fever during early childhood (OR 1.48 (95% CI, 1.18–1.84), *p* = 0.0005) were associated with MIH. In conclusion, the etiology of MIH was found to be multifactorial. Children with health disorders in the first years of life and those whose mothers underwent illnesses during pregnancy might be more susceptible to MIH.

## 1. Introduction

Molar incisor hypomineralization (MIH) is a qualitative developmental defect of the dental enamel that affects one or more of the permanent first molars and, occasionally, the incisors of the same dentition. Clinically, it is characterized by marked opacities that vary in hue from whitish to yellow/brown. Their degrees of severity are distributed asymmetrically, and due to the fragility of the affected areas, fractures can occur once they are exposed to the forces of mastication [1] The criteria for MIH diagnosis were initially described by the European Academy of Pediatric Dentistry [2], and, subsequently, modifications were proposed based on the disease severity in order to include other dental groups [3]. MIH injuries can occur following the dental eruption of the first molars and permanent incisors. In addition, a predictive factor for MIH can be the presence of hypomineralized second primary molars. However, there have been reports of cases where the absence of this defect in the deciduous dentition was unable to rule out future MIH occurrence [4]. Furthermore, the probability of the second primary molars, permanent canines, and premolars showing signs of enamel hypomineralization increases when the molars and incisors are affected [5].

The prevalence of MIH was estimated to be 13.5%, while moderate-to-severe cases of MIH were estimated at 36.3% [6]. Dental care for MIH is complex because of disorders in the morphology and structure of the enamel prisms. Additionally, due to the considerable porosity and fragility resulting from this condition, sufferers have a great susceptibility to caries and hypersensitivity. This hypersensitivity is caused by an inflammatory reaction of the pulp that causes changes in the sensory neurons where, even with local anesthesia, the teeth present discomfort to cold or painful stimuli during dental treatment. This discomfort could alter the behavior of pediatric patients [7]. Additionally, the adhesion of materials on the teeth with MIH makes restorative treatments difficult, which may lead to recurrences and repetitive interventions, ultimately leading to tooth extraction.

MIH occurs during the enamel’s apposition and maturation phase, and despite multiple reports on this topic, its etiology and pathogenesis remain unclear [8]. Considering that disturbances in the mineralization of MIH can alter the quality of life of a child, an analysis of recently published scientific evidence on MIH-predisposing factors is needed in order to improve the existing measures for more effective MIH prevention. Thus, we conducted a qualitative and quantitative analysis of the recent literature to identify the significant risk factors associated with MIH occurrence.

## 2. Materials and Methods

The protocol for this systematic review was designed by all of the authors. This study was registered at the National Institute for Health Research PROSPERO, International Prospective Register of Systematic Review (ID: CRD42022348603), and was designed following the preferred reporting items for systematic review and meta-analysis (PRISMA) guidelines. [9,10]

### 2.1. Search Strategy and Study Selection

The relevant literature was systematically extracted from six international databases, including PubMed, Web of Science, Scopus, the Latin American and Caribbean Health Sciences Literature database (LILACS), and SciELO, from the period between 2004 and 17 December 2022. Medical subject headings (MeSH) and free terms were combined according to the syntax rules for each database. Terms related to MIH and risk factors were searched. Our search strategy included the following terms: “hypomineralization molar AND factor risk” or “hipomineralización molar”. Moreover, we manually screened for potentially relevant publications from the references of our retrieved studies. The above process was performed independently by two participants (BHM and LVST).

### 2.2. Participants and Eligibility Criteria

The study’s inclusion criteria, including PECO, were as follows: (1) Patient: MIH children below the age of 18 years old. (2) Exposure intervention: We included studies which focused on the possible association between MIH and prenatal, perinatal, and postnatal problems. (3) Comparison: participants with no risk exposure or an absence of systemic exposure. (4) Outcome indicator: Prenatal factors (illness/infections during pregnancy), perinatal factors (birth weight, premature birth, or cesarean section), and postnatal factors (childhood illness, antibiotic use, and high fever) were taken into account as risk factors by the studies [11]. The odds ratios (OR) with 95% confidence intervals (CI) for each factor were considered or computed if enough relevant data were available. (5) Types of studies: We included studies properly registered and approved by their relevant ethics committees: case–control, cross-sectional, and cohort studies, as well as clinical trials, were utilized. Only studies in English, Spanish, and Portuguese were included.

Studies were excluded if: (1) they were case reports, reviews, summaries of discussions, or in vitro studies; (2) they contained insufficient data for analysis; (3) the patients were stratified based on the degree of severity; (4) they involved genetic factors; or (5) the patients were above 18 years old.

### 2.3. Data Extraction and Quality Assessment

The literature screening, data extraction, and literature quality evaluation were conducted separately by two analysts (BHM and LVST). Any differences were resolved through mutual discussion or following consultation with a third analyst (MLAJL).

A data extraction spreadsheet was designed, and the following information was independently extracted by two reviewers (LVST and BHM): first author’s surname, year of publication, country, participant characteristics, and statistical summaries related to factor risk (prenatal, perinatal, and postnatal etiological factors).

The quality of each of the included studies was assessed independently by the two analysts, and the Newcastle–Ottawa scale (NOS) was used to evaluate the process in terms of queue selection, the comparability of queues, and the evaluation of results to obtain a final score [12]. The NOS is a star rating system that assigns a maximum of nine stars across three categories. High quality was attributed to studies that scored three or four stars in selection, one or two stars in comparability, and two or three stars in outcome/exposure. Studies with scores of two stars in selection, one or two stars in comparability, and two or three stars in outcome/exposure were considered to be of moderate quality. When the studies scored no stars or one star in selection, no stars in comparability, and no stars or one star in outcome/exposure, they were considered to be of low quality. In this review, 40 studies with scores ≥ 5 were included for analysis.

### 2.4. Analysis and Meta-Analyses

A standardized data extraction form was used for qualitative analyses to record the study characteristics (author and publication year), design, and the main findings of etiological factors. Meta-analyses and heterogeneity calculations were conducted using RevMan (version 5.4). The odds ratio (OR) was calculated using a 95% confidence interval (CI) to test the association between MIH and etiological factors. Only articles that had OR data were considered for the meta-analyses. An I^2^ statistical analysis was also performed, with I^2^ values ≤ 25%, 25–50%, 50–75%, and >75% indicating no, low, moderate, and significant heterogeneity, respectively. Forest plots were constructed for maternal illnesses; prematurity; low weight; cesarean birth; and postnatal factors such as respiratory problems, general illnesses, intake of antibiotics, and the presence of fever in childhood. A *p*-value ≤ 0.05 was considered statistically significant.

## 3. Results

Our database search initially identified 367 potential articles from the 6 databases. After the exclusion of duplicates, 271 articles were considered for further screening. Of these, 222 studies were excluded after screening their titles and abstracts. After a careful review of the full texts of the remaining articles, 40 studies were selected for qualitative analysis. Then, 15 articles were excluded because they had insufficient data or lacked the variable of interest. Finally, a total of 25 articles that met the inclusion criteria and were of high quality were included in this meta-analysis. The flow diagram in Figure 1 shows the detailed literature search of the present study.

In the qualitative analysis, 95% of the 40 studies used the European Academy of Pediatric Dentistry criteria for diagnosing MIH. The studies were from Asia (17), Europe (10), and America (13). The main findings of the included studies are presented in Table 1. There were 28 cross-sectional studies [13,14,15,16,17,18,19,20,21,22,23,24,25,26,27,28,29,30,31,32,33,34,35,36,37,38,39,40] in which a questionnaire on medical history was used in the prenatal and perinatal stages and in the first 3 years of life; 5 were cohort studies [41,42,43,44,45]; and 7 were case–control studies [46,47,48,49,50,51,52]. The prevalence of MIH ranged from 2.5 to 54% in 44,037 participants. The Newcastle–Ottawa (NOS) scale values, shown in Table 1, showed that 75% (30) of the studies were of high quality and 25% (10) were of moderate quality; regarding the risk of bias, 33 corresponded to a moderate degree of risk and 7 to a low risk.

Twenty-five studies were used for the quantitative analysis. Since the heterogeneity test showed significant heterogeneity among all the studies, the random effect model was used for all of them.

Table 2 illustrates the main etiological factors of this meta-analysis. At the prenatal stage, the results showed that a history of diseases during the last trimester of pregnancy was a factor risk for MIH (OR 4.03 [95% CI, 1.33–12.16]; *p* = 0.01) (Figure 2).

Regarding the perinatal stage, the results for prematurity and birth by cesarean section did not show any statistical significance and demonstrated a heterogeneity of 77%. Ten studies were included for low-birth-weight analysis and demonstrated an OR of 1.23 (95% CI, 1.10–1.38, *p* = 0.0005) (Figure 2). Preterm birth and cesarean birth were analyzed separately; no significant association with MIH was observed (*p* > 0.05).

At the postnatal stage, alterations in health during the first years of life correlated well with MIH; Figure 3 presents the forest plot for the association between MIH and history of diseases, antibiotics use, and high fever during the postnatal period, showing an OR of 4.06 (95% CI, 2.03–8.11; *p* = 0.00001) and an I^2^ of 94% for childhood illness. In the case of long-term antibiotic use in the first years of life, the OR was 1.76 (95% CI, 1.31–2.37) and the I^2^ was 83%. Lastly, for fever, the meta-analysis showed an OR of 1.48 (95% CI, 1.18∓1.84; *p* = 0.0005) and an I^2^ of 54%. In this analysis, the relationship with respiratory disorders was not confirmed for MIH.

## 4. Discussion

This systematic review discussed the factors associated with MIH. In the analysis of environmental factors related to MIH in the prenatal period, Souza et al. [16], Ghanin et al. [17], Koruyucu et al. [25], Mejía et al. [28], Mariam et al. [37], and Verma et al. [38] reported an association with health problems during pregnancy. Rai et al. highlighted an association with diabetes, hypertension, and vitamin D deficiency in the last three months of pregnancy [24]. Therefore, the pooled results of this meta-analysis coincided with the findings of Fattury et al., whose systematic review indicated that mothers with health problems might have a 40% higher risk of having children with this alteration in their enamel structure [53]. A meta-analysis by Garot et al. [54] also reported an association with maternal illness. Another systematic review by Silva et al. did not reveal an association, and reported variability in the terms used to describe maternal illness. Any alteration during amelogenesis could cause MIH, so there are different diseases during pregnancy that could be determinants for the oral health and quality of life of children [55].

A poorly studied risk factor for MIH is exposure to contaminants from consuming canned or packaged food and beverages in plastics during pregnancy and the first years of life. In this regard, Glodlowska et al. [30] and Elzein et al. [34] reported an association, and other studies have noted that bisphenols could cause hormonal disturbances and increases in proteins such as enamelin and albumin during amelogenesis, which could lead to alterations in the formation and maturation of the enamel [56,57].

On the other hand, regarding the perinatal period, it is known that complications during birth can cause suffering in the child and deficiencies in oxygenation that affect amelogenesis [19]. In this meta-analysis, we did not find an association between MIH and prematurity or cesarean birth, which might be due to the low OR values of the included studies, which might have in turn decreased the strength of the association. In addition, our results are similar to those of other systematic reviews [55,58]. Low birth weight is a significant problem worldwide, and can lead to illnesses in both the short and long term. It has been associated with low socioeconomic status and diseases of the pregnant mother, which partly explains the potential predictive ability of maternal illness and low weight for the MIH occurrence observed in this work. These findings support the observation of an umbrella study which mentioned that a simultaneous combination of two or more risk factors is required for MIH to occur [59].

The finding regarding childhood diseases during the first years of life in the etiology of MIH agrees with that of Fatturi et al. [53] and Garot et al. [54], corroborating the association with MIH.

Most of the included studies showed a relationship of MIH with allergies; asthma; ear infectious processes; and urinary and/or respiratory diseases, including adenoiditis, tonsillitis, and pneumonia. Other works have found an association with exposure to pollutants [30,34]. However, it should be noted that in a meta-analysis that exclusively included respiratory disorders of the postnatal stage, the association with MIH was imprecise; likewise, the case–control study by Nogueira et al. [47], as well as the cohort study by Van der Tas et al., also did not find evidence for their relationship [43]. On the other hand, regarding exposure to medications in early childhood, there is not enough evidence about corticosteroids and antineoplastic drugs. In the present work, antibiotics, one of the most frequently used medications in childhood, were found to be related to both MHI and high fever. Similarly, a study mentioned that amoxicillin might alter the expression of some genes essential for enamel development [60], and reported that high fever altered the stage of enamel matrix formation and enamel mineralization [61]. However, it remains to be precisely defined whether the alteration in ameloblastic function is due to illness in childhood, related fevers, or the administered medication. Additionally, conclusive evidence also could not be obtained with in vitro studies [62,63,64], suggesting the need for further research.

In recent years, there has been evidence of a codominant inheritance complex for MIH. Therefore, it is probable that genetic susceptibility and genes might be involved in its phenotype. It has been suggested that the susceptibility to developing MIH is increased by exposure to risk factors such as medication use during amelogenesis [65,66]. Furthermore, although there seems to be an association between genetic polymorphism in enamel formation and MIH, more research is needed for clarification [67].

Moreover, since teeth with HIM have a high risk of dental caries due to their poor mineralization structure, rapid breakdown, and hypersensitivity symptoms [6], using toothpastes with biometric zinc hydroxyapatite and desensitizers such as arginine and calcium carbonate would be recommended. It is reported that due to its physical characteristics and ionic substitution capacity, biometric zinc hydroxyapatite remineralizes enamel and dentin, blocking the dentinal tubules and reducing the hypersensitivity present in MIH cases, thus improving pediatric treatment and preventing the recurrence of carious lesions [68,69,70].

Additionally, agents that contain casein phosphopeptide amorphous calcium fluoride phosphate (CPP-ACFP) and calcium glycerophosphate (CaGP) were demonstrated to be new, noninvasive, patient-friendly, and effective tools to treat reversible carious lesions in teeth affected by MIH [71].

On the other hand, there is controversy surrounding the association between MIH and calcium and vitamin D deficiencies during pregnancy and the first years of life [24,43]; as a safe prevention strategy, it is advisable to prescribe multivitamins, calcium, and folic acid to pregnant women to prevent MIH and any other structural enamel alterations. In this regard, Kunish pointed out that high concentrations of vitamin D favored better oral health conditions [72]. Prenatal supplementation could be beneficial in preventing enamel defects, including MIH [73].

The limitations of this study include the language of the screening, which may have limited the exhaustiveness of the review. Despite this, this work provides a summary of the recent evidence on MIH risk factors. The medical problems of pregnant mothers and children were analyzed in general. It would be interesting to perform a meta-analysis for each disease; eliminating those studies with high variability would allow us to obtain more precise results. The results presented in this systematic review suggest that the occurrence of MIH is due to a combination of multiple etiologies. It is important to note that most of the included studies were cross-sectional, based on questionnaires that inquired about the medical history of the child and mother. Additionally, the number of studies was low. Thus, the results reported in this study should be considered with caution. To obtain more convincing results on the etiopathogenesis of MIH, it is important that future studies be based on well-documented medical records, cases, and controls, and that they incorporate genetic variability. It would be useful to carry out medical/dental cohort studies that show the follow-up of cases exposed to prenatal and perinatal factors for MIH, as well as to consider publications in other languages.

Furthermore, disseminating information among physicians, pediatricians, and parents about the predisposing factors could prevent oral health complications and reduce the effects on the quality of life of patients with MIH.

Additionally, it is important to implement a care program for pregnant women and children exposed to risk factors (pre-, peri-, and postnatal stages) in which physicians provide timely dental follow-ups for early diagnosis, especially in the eruption stage.

Finally, it is important that medical staff are encouraged to become informed about the prescription of antibiotics. It is also essential that guidance be provided on the possible effects of polluting products consumed by pregnant mothers and their children.

## 5. Conclusions

MIH depends on a multifactorial etiology, where exposure to various combined factors with genetic susceptibility determines the presence and severity of the disease, as well as the affected dental group.

This study indicates that illnesses during pregnancy, low birth weight, and illnesses during the first years of life could be related to the occurrence of MIH. Additionally, high fever and antibiotic use in early childhood could also be potential risk factors.

All the strategies aimed at healthcare for pregnant women and children during the first years of life are relevant for preventing MIH. The dental follow-up of children who are exposed to risks during the pre-, peri-, and postnatal stages is highly recommended for the timely diagnosis and treatment of dental disorders such as MIH.

## Figures and Tables

**Figure 1 dentistry-11-00111-f001:**
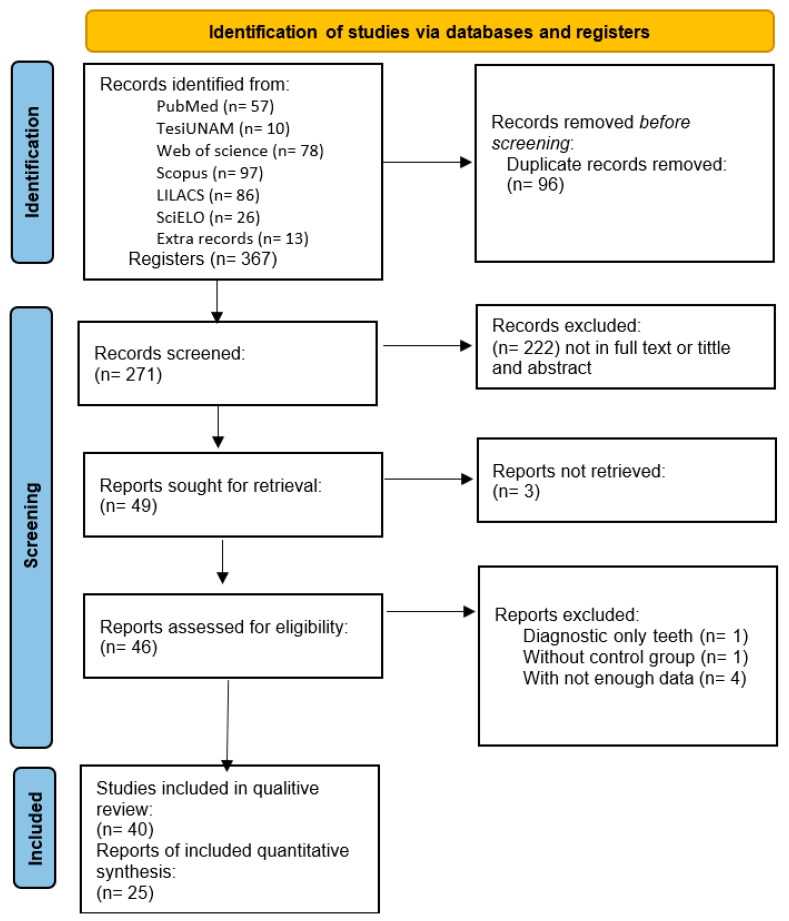
Flow diagram summarizing study identification and selection.

**Figure 2 dentistry-11-00111-f002:**
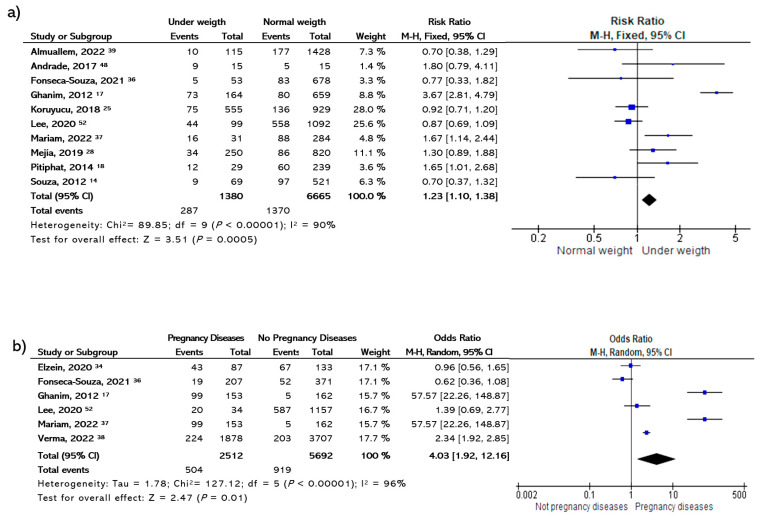
Forest plot showing the association of (**a**) low birth weight and (**b**) illnesses during pregnancy with the risk of MHI [14,17,18,25,28,34,36,37,38,39,48,52].

**Figure 3 dentistry-11-00111-f003:**
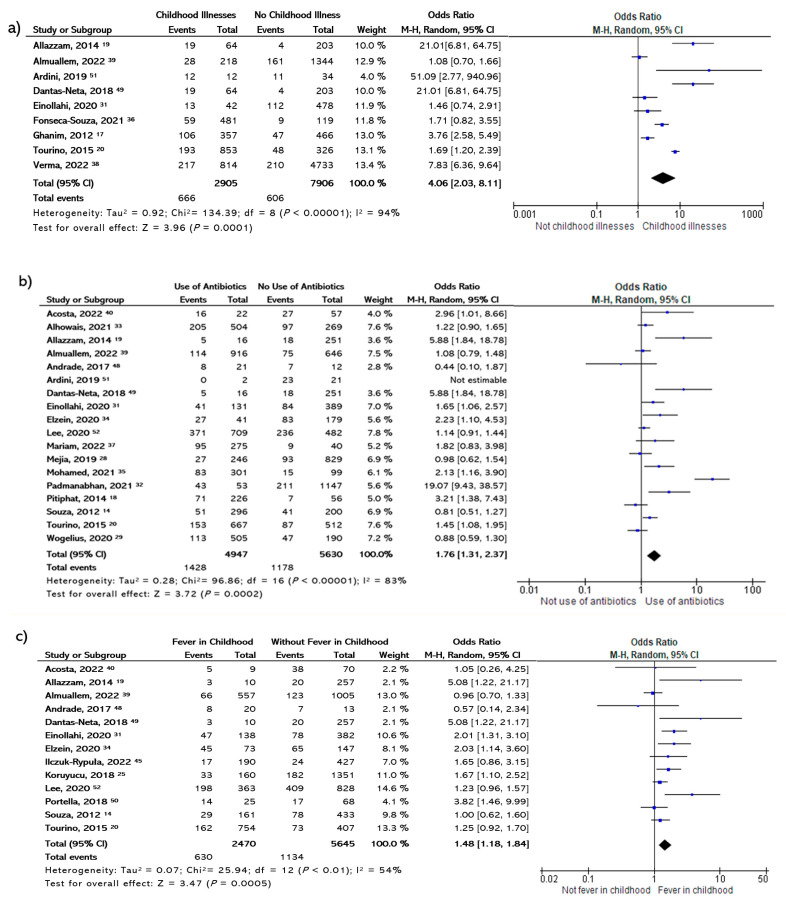
Forest plot showing the association between different statuses in childhood and risk of MHI, including (**a**) illnesses, (**b**) long-term use of antibiotics, and (**c**) fever [14,17,18,19,20,25,28,29,31,32,33,34,36,37,39,40,45,48,49,50,51,52].

**Table 1 dentistry-11-00111-t001:** Type of study, prevalence, Newcastle–Ottawa scale scores, and main findings of the selected studies. S = selection; C = comparability; and E = exposure.

Author Names and Country	Type of Study	No. of Participants	MIH Prevalence (%)	Newcastle–Ottawa			*Findings*
				S	C	E	
Brogard-Roth et al., 2010 [13] Sweden	Cross- sectional case–control study	144	38	***	*	*	*Low gestational age and low birth weight increased the risk of MIH and oral health problems (more plaque, gingival inflammation, and behavior management problems).*
Bondi et al., 2010 [46] Argentine	Case–control	196	50	***	**	*	*MIH presented an association with respiratory infections, special formula milk, and nonsteroidal and anti-inflammatory drugs.*
Souza JF. et al., 2012 [14] Brazil	Cross-sectional	903	19.9	***	*	***	*Health problems in pregnancy. History of throat infections, periods of fever, and amoxicillin intake in the postnatal stage were found to be associated with MIH. There was no correlation with birth prematurity or low weight at birth.*
Sönmez H. et al., 2012 [15] Turkey	Cross-sectional	3827	7.7	***	**	***	*Prematurity, gastrointestinal problems, pneumonia, frequent fever, measles, and chickenpox before 4 years old were found to be associated with MIH.*
Souza JF. et al., 2013 [16] Brazil	Cross-sectional	1151	12.3	***	**	**	*Cesarean birth, low weight, and maternal illness were not associated with MIH.* *The occurrence of anemia was not significant.*
Ghanim A. et al., 2013 [17] Iraq	Cross-sectional	823	18.6	***	*	***	*Infections and lung diseases, unexplained fever, tonsillitis, pneumonia, exposure to drugs during the first year, and breastfeeding for less than 6 months were found to be associated with MIH.*
Kühnisch J. et al., 2013 [41] Germany	Cohort	693	37.9	**	*	***	*There was an association* *with respiratory problems.*
Pitiphat W. et al., 2014 [18] Thailand	Cross-sectional	282	27.7	***	*	***	*Birth by cesarean section and poor health in the first three years were found to be associated with MIH.*
Allazzam SM. et al., 2014 [19] Egypt	Cross-sectional	267	8.6	***	*	**	*Many diseases during the first four years of life (tonsillitis, adenoiditis, and asthma), as well as antibiotic consumption and febrile symptoms, were found to be associated with MIH.*
Nogueira F. et al., 2015 [47] Brazil	Case–control	1237	2.5	**	*	**	*HMI has no association with prenatal or postnatal factors in the first three years of life.*
Tourino L. et al., 2015 [20] Brazil	Cross-sectional	1181	20.4	***	**	**	*Asthma/bronchitis, high fever, and the use of antibiotics in the first four years of life were factors associated with MIH.*
Woullet E. et al., 2016 [42] Finland	Cohort	287	11.5	****	*	***	*Acute otitis media and the use of penicillin and macrolides were found to be associated with MIH.*
Garot E. et al., 2016 [21] France	Cross-sectional	819	8	**	*	**	*Hypoxia during birth and birth by cesarean section were found to be associated with MIH.*
Barbosa T. et al., 2017 [22] Brazil	Cross-sectional	167	29.3	***	*	*	*There was a genetic influence on the occurrence of MIH; preterm birth was not related to MIH.*
Andrade NS. et al., 2017 [48] Brazil	Case– control	99	45.5	***	**	**	*The prevalence of MIH and dental caries was increased in children and adolescents infected with HIV.*
Gurrusquieta BJ. et al., 2017 [23] Mexico	Cross-sectional	1156	182	***	*	**	*MIH was found to be associated with low weight, urinary tract infections, and allergies in the first years of life.*
Rai A, et al., 2018 [24] India	Cross-sectional	992	21.4	***	*	**	*Vitamin D deficiency, diabetes, or hypertension in pregnancy;* *prematurity or complications during delivery; and infections in the early years were found to be associated with MIH.*
Koruyucu M. et al., 2018 [25] Turkey	Cross-sectional	1511	14.2	***	*	**	*MIH was found to be associated with birth prematurity, diarrhea frequency, digestive system diseases, renal failure, rubeola, and chickenpox in the early years of life.*
Van der Tas JT. et al., 2018 [43] Netherlands	Cohort	3406	8.1	****	**	***	*There was no association with calcium concentrations in the prenatal or postnatal stages.*
Giuca M. et al., 2018 [26] Italy	Cross-sectional	120	50	**	*	**	*Many infections (ear, throat, and nose) and the ingestion of antibiotics in the first years of life were risk* *factors for MIH.*
Dantas-Neta NB et al., 2018 [49] Brazil	Case– control	744	19.5	***	**	*	*MIH was found to be associated with the presence of fever during gestation.*
Portella PD. et al., 2018 [50] Brazil	Case–control	93	-	**	*	**	*MIH was associated with prematurity and prolonged delivery. In addition, in the postnatal period, recurrent fevers in the first 3 years of life were associated with MIH.*
Kılınç G. et al., 2019 [27] Turkey	Cross-sectional	1237	11.5	***	*	**	*Preterm delivery, bronchitis, asthma, and high fever in early childhood were found to be associated with MIH.*
Ardini Y. et al., 2019 [51] Malaysia	Case– control	156	14.3	**	*	**	*Childhood illness, but not with perinatal complications or prolonged antibiotics consumption, was associated with MIH.*
Mejía JD. et al., 2019 [28] Colombia	Cross-sectional	1075	11.2	**	*	**	*Alterations during the last gestational trimester, premature delivery, maternal illness or infection, and/or maternal hypocalcemia and respiratory diseases were associated with MIH.*
Woegelius P. et al., 2020 [29] Denmark	Cross-sectional	1837	29.5	****	**	**	*There was no association between the use of inhaled asthma medication and MIH.*
Flexeder C. et al., 2020 [44] Germany	Cohort	750	37.5	****	*	**	*There was an association between asthma without medication and MIH.*
Głódkowska N. et al., 2020 [30] Poland	Cross-sectional	2275	9.32	**	**	**	*Exposure to higher concentrations of air pollutants and respiratory illnesses were found to be associated with MIH.*
Einohalli M. et al., 2020 [31] Irak	Cross-sectional	520	24%	***	*	*	*Asthma (or bronchitis), hospitalization history, and fever (above 38.5 °C) were factors associated with MIH.*
Lee DW. et al., 2020 [52] Korea	Case–control	1191	50	***	**	**	*Maternal smoking during pregnancy and pediatric respiratory infection (suffered in early childhood) could predict MIH.*
Padmanabhan V. et al., 2021 [32] United Arab Emirates	Cross-sectional	1200	21.6	**	**	*	*Early-childhood illnesses (adenoiditis, tonsillitis, and asthma) were factors associated with MIH. However, high fever was not* *significantly associated with perinatal variables.*
Alhowaish L. et al., 2021 [33] Saudi Arabia	Cross-sectional	893	40.5	***	*	**	*Newborn jaundice was a factor associated with MIH.*
Elzein R. et al., 2021 [34] Lebanon	Cross-sectional	659	26	***	*	**	*Otitis media, fever, antibiotics, and the consumption of canned foods and beverages in the early years could predict MIH. Medical problems during pregnancy and mother’s medication during feeding were not significantly associated.*
Mohamed RN. et al., 2021 [35] Saudi Arabia.	Cross- sectional.	400	24.5	****	*	***	*Children with a breastfeeding history >18 months had a greater risk of MIH.*
Fonseca G. et al., 2021 [36] Brazil	Cross-sectional	731	12.10	***	**	**	*Prematurity, prolonged delivery, and recurrent fevers could predict MIH.*
Mariam S. et al., 2022 [37] India	Cross-sectional	3176	11.72	****	**	**	*Maternal anemia, preterm and low-weight birth, neonatal problems, early-childhood illnesses, medication in the first years, and socioeconomic status were associated with MIH.*
Verma S. et al., 2022 [38] India	Cross- sectional	5585	7.6%	***	**	**	*Mother’s illness and the intake of medications during pregnancy and by infants in the initial 4 years of life were associated with MIH.*
Almuallem Z. et al., 2022 [39] Saudi Arabia	Cross- sectional	1562	15.2	***	*	**	*Childhood illness (ear infections, respiratory distress, and tonsillitis) during the first three years of life showed a strong positive association with MIH.*
Acosta E. et al., 2022 [40] Spain	Cross-sectional	79	54.43	**	*	**	*MIH was significantly associated with the administration of Haloperidol during delivery. Additionally, serious infections and antibiotics in the first years of life could predict MIH.*
Ilczuk-Rypuła D. et al., 2022 [45] Poland	Cohort	613	6.2	***	**	*	*Otitis in early childhood, atopic dermatitis, and preterm birth before 38 weeks of pregnancy were significantly associated with MIH.*

* ** *** ****: The NOS is a star rating system. Stars are awarded such that each category can awarded up to four stars when your quality is high. S = selection; C = comparability; E = exposure.

**Table 2 dentistry-11-00111-t002:** Meta-analyses with main etiological factors of MIH.

Etiologies	N Study	N Participants	Chi^2^	I^2^	OR (95%IC)	*p*-Value
Prenatal						
Disease in pregnancy	6	5692	127.12	96	4.03(1.33–12.16)	0.01 *
Perinatal						
Premature birth	18	9355	62.16	77	1.26(0.99–1.59)	0.06
Low weight	10	6665	89.85	90	1.23(1.10–1.38)	0.0005 *
Cesarean section	12	4017	43.06	77	0.77(0.59–1.00)	0.05
Postnatal						
Respiratory diseases	12	7455	249.5	96	1.51(0.79–2.37)	0.21
Antibiotic consumption	18	5630	96.86	83	1.76 (1.31–2.37)	0.0002 *
General childhood illnesses	9	7906	134.39	94	4.06(2.03–8.11)	0.0001 *
High fever	13	5645	25.94	54	1.48(1.18–1.84)	0.0005 *

* Statistically significant.

## Data Availability

All of the data supporting the conclusions of this article are contained within the manuscript. The study dataset can made available by the authors upon reasonable request.

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
