# Peer review of "Etiological Factors of Molar Incisor Hypomineralization: A Systematic Review and Meta-Analysis"

_dentistry, 2023, doi:10.3390/dj11050111_

Round 1

Reviewer 1 Report

Modifications Needed:

- Review the wording of the first sentence of the discussion

- The quality of figures 2 and 3 is poor

- Some citations in the text do not match the bibliography

- There are incomplete bibliographic citations

- There are bibliographic citations with formatting errors

Author Response

We appreciate your kind comments and according to your suggestions

Point 1.  Review the wording of the first sentence of the discussion.

 It was corrected

Point 2. The quality of figures 2 and 3 is poor.

The figures 2 and 3 were modified to get more clarity

Point 3. Some citations in the text do not match the bibliography.

 We review and correct the bibliography

Point 4.  There are incomplete bibliographic citations

 The bibliography references were reviewed and corrected

Point 5. There are bibliographic citations with formatting errors

 We review and correct the bibliography

Reviewer 2 Report

Dear Authors, 

Here is the revision of your work entitled “Etiological factors of molar incisor hypomineralization: A systematic review and meta-analysis”. There are some issues to be solved in order to improve the manuscript for publication. Please, highlight the corrections with a color mark and provide a point-by-point response.

Abstract: The purpose of this review ïƒ  The purpose of this systematic review

Keywords: you have up to 10 keywords. You can add more of them.

Introduction

-       “Additionally, due to its considerable porosity and fragility, it has a great susceptibility to caries and hypersensitivity. The adhesion of materials makes restorative treatments difficult, which may lead to recurrences and repetitive interventions, ultimately leading to tooth extraction.” It is suggested to add that the problem of hypersensitivity is so troubling that intraoperative pain after local anesthesia increases for behavioral issues (DOI: 10.17796/1053-4625-46.3.6).

Materials and Methods: it should be clearly specified the language in which articles were selected. 

Results

-       The quality of figure 1 should be improved. Please, try to export the original file with higher quality. 

-       The same can be said for figure 2 and figure 3. Moreover, add the citation reference at the end of the column Study group (e.g. Almuallem, 2022 [39]). If possible, figure 2 and figure 3 should be presented as tables, so that the readers can use the search tool for the text. 

-       Table 1 heading: please, add the legend of S, C, E letters (selection, comparability, Exposure). 

Discussion

-       I suggest dividing the Discussion and the results sections into subsections according to the variable analysed. This will help the readers to better understand.

-       It should be noted that understanding the etiological factors of MIH is surely of great interest, but it does not add relevant information on how to prevent its onset. In fact, the problems of hypersensitivity and fragile enamel remains. This issue should be properly discussed. 

-       It is suggested improving the last paragraph with more details on treatment for MIH, here are some recent trials which can be taken as examples (DOI: 10.17796/1053-4625-46.3.4; DOI: 10.1111/odi.14388).

Conclusion: “However, due to some of the limitations in this study, more research is needed to increase our understanding on MIH.” This sentence should be removed. The limitations of the study should be discussed in the Discussion section. 

General issues

-       English grammar and punctuation should be extensively revised. 

-       Personal writing is not desirable, please rephase in impersonal writing the entire manuscript. 

Author Response

We are grateful for reading and comments this work, we really appreciate your expertise and time.

Point 1. Abstract: The purpose of this review à The purpose of this systematic review

We completed the sentence with systematic word

Point 2. Keywords: you have up to 10 keywords. You can add more of them.

Modifications were made considering your wise comments in key words.

Point 3, Introduction

-   Additionally, due to its considerable porosity and fragility, it has a great susceptibility to caries and hypersensitivity. The adhesion of materials makes restorative treatments difficult, which may lead to recurrences and repetitive interventions, ultimately leading to tooth extraction.” It is suggested to add that the problem of hypersensitivity is so troubling that intraoperative pain after local anesthesia increases for behavioral issues (DOI: 10.17796/1053-4625-46.3.6).

According your comments for the introduction text, we incorporate current aspects for the hypersensitivity of MIH

Point 4. Materials and Methods: it should be clearly specified the language in which articles were selected. 

We added the languages which articles were selected

 Point 5. Results-       The quality of figure 1 should be improved. Please, try to export the original file with higher quality. 

The quality of figure 1 was improvedT

Point 6, -       The same can be said for figure 2 and figure 3. Moreover, add the citation reference at the end of the column Study group (e.g. Almuallem, 2022 [39]). If possible, figure 2 and figure 3 should be presented as tables, so that the readers can use the search tool for the text. 

We added reference at the end of the column of the authors in the figures 2 and 3.

Point 7. -       Table 1 heading: please, add the legend of S, C, E letters (selection, comparability, Exposure). 

We added the legend

Point 8. Discussion

-       I suggest dividing the Discussion and the results sections into subsections according to the variable analysed. This will help the readers to better understand.

We reorganized results and discussion

Point 9. Discusion

-       It should be noted that understanding the etiological factors of MIH is surely of great interest, but it does not add relevant information on how to prevent its onset. In fact, the problems of hypersensitivity and fragile enamel remains. This issue should be properly discussed. -       It is suggested improving the last paragraph with more details on treatment for MIH, here are some recent trials which can be taken as examples (DOI: 10.17796/1053-4625-46.3.4; DOI: 10.1111/odi.14388).

According your comments for the discussion text, we incorporated current aspects for prevention and treatment  of MIH

Point 9. Conclusion: “However, due to some of the limitations in this study, more research is needed to increase our understanding on MIH.” This sentence should be removed. The limitations of the study should be discussed in the Discussion section. 

-We modified the wording of conclusions and reviewed the entire manuscript

Point 10. General issues

-       English grammar and punctuation should be extensively revised. 

-       Personal writing is not desirable, please rephase in impersonal writing the entire manuscript.

-We reviewed the wording of the entire manuscript and the manuscript has been  reviewed by an English language editor as you suggested

kindly regards

Reviewer 3 Report

Dentistry Journal – MDIP

Etiological factors of molar incisor hypomineralization: A systematic review and meta-analysis 

Reviewer’s Comments to Author and Editor: 

This systematic review paper on molar incisor hypo mineralization is well-written and provides a valuable resource for clinicians and researchers alike. The authors conducted a good review of the topic and presented the findings in a clear and concise way, with well-presented tables and figures. However, I would have appreciated a more detailed explanation about the quality of the studies selected and the presence of bias, as these are important factors in a systematic review.

The number of papers identified through six databases (367) seems somewhat small for an initial search, which may have limited the comprehensiveness of the review.

Additionally, the last paragraph of the conclusion suggests the use of calcium and folic acid to prevent MIH, but this recommendation is not supported by the review and is not referenced.

Finally, while the authors mention limitations in the conclusion section, I suggest adding a paragraph at the end of the discussion that specifically outlines the limitations of the study and how they may have impacted the findings. 

Author Response

We appreciate your comments and have made the follow modifications to improve the manuscript :

Point 1. I would have appreciated a more detailed explanation about the quality of the studies selected and the presence of bias, as these are important factors in a systematic review

We  complemented the criteria for quality and bias applied in methodology section, and commented about studies in results section

Point 2. The number of papers identified through six databases (367) seems somewhat small for an initial search, which may have limited the comprehensiveness of the review.

-At the database searching, when we considered the MIHS terms and the language limitation in period selected from 2004 to 2022, so 367 publications on MIH risk factors were obtained, which were those that were considered at eligibility step. We commented at discussion section as limitation of the review

Point 3. Additionally, the last paragraph of the conclusion suggests the use of calcium and folic acid to prevent MIH, but this recommendation is not supported by the review and is not referenced.

The use of vitamins are good for oral health and to prevent enamel defects, so we added references that support this recommendation.

Point 4, Finally, while the authors mention limitations in the conclusion section, I suggest adding a paragraph at the end of the discussion that specifically outlines the limitations of the study and how they may have impacted the findings.

Thanks, we added at the end of the discussion the limitations of the study

Reviewer 4 Report

The authors had a study on etiological factors associated with MIH..

The introduction section is good and provides sufficient background and includes all relevant references. However, I think some recent references are needed to add.

Please mention if there was a limitation on language in the search strategy.

Please mention if the authors used MESH keywords.

How the data was screened?

Please improve the quality of figure 1.

Please add the reference numbers in Table 1.

Please improve the quality of figure 1.

The conclusion section is short and not supported the results. Please rewrite it totally.

Please add some more recent references.

Please check the typo errors.

Author Response

We appreciate your comments and have made the follow modifications to improve the manuscript

Point 1 The introduction section is good and provides sufficient background and includes all relevant references. However, I think some recent references are needed to add.

-In introduction, we included recent references and information about hypersensitivity in patients with MIH.

Point 2. Please mention if there was a limitation on language in the search strategy.

We complemented that limitation in the search strategy

Point 3, Please mention if the authors used MESH keywords.

Thanks , we did the mention

Point 4, How the data was screened?

The articles were analyzed by the authors to extract the relevant data from their findings. the prevalence and main findings are presented  in table 1

Point 5. Please improve the quality of figure 1.

We are sending figure 1 and 2,  with a better quality

Point 6, Please add the reference numbers in Table 1.

We added reference at the end of the column of the authors in the figures 2 and 3.

Point 7. The conclusion section is short and not supported the results. Please rewrite it totally.

We modified the wording of conclusions

Point 9. Please add some more recent references.

We added some recent references in introduction and discussion

Point 10. Please check the typo errors.

-We reviewed the wording of the entire manuscript and the manuscript has been  reviewed by an English language editor as you suggested

Round 2

Reviewer 2 Report

Dear Authors, 

Thank you for providing the revised version of your manuscript. Now it is suitable for publication. 

Reviewer 4 Report

The modifications are good. It can be accepted.